



# Thermospheric nitric oxide NO during solar minimum modulated by O/O$_2$ ratio and thermospheric transport and mixing

Miriam Sinnhuber[1], Christina Arras[2,9], Stefan Bender[3], Bernd Funke[3], Hanli Liu[4], Daniel R. Marsh[5], Thomas Reddmann[1], Eugene Rozanov[6], Timofey Sukhodolov[6], Monika E. Szelag[7], and Jan Maik Wissing[8]

[1]Institute of Meteorology and Climate Research, Karlsruhe Institute of Technology (KIT), Karlsruhe, Germany
[2]Space Geodetic Techniques, German Research Centre for Geosciences GFZ, Potsdam, Germany
[3]Instituto de Astrofisica de Andalucia, CSIC, Granada, Spain
[4]High Altitude Observatory, National Center for Atmospheric Research NCAR, Boulder, US
[5]School of Physics and Astronomy, University of Leeds, Leeds, UK
[6]Physikalisch-Meteorologisches Observatorium Davos and World Radiation Center, Davos, Switzerland
[7]Finish Meteorological Institute FMI, Helsinki, Finland
[8]Institute for Solar-Terrestrial Physics, Deutsches Zentrum für Luft- und Raumfahrt DLR, Neustrelitz, Germany
[9]Technische Universität Berlin, Berlin, Germany

**Correspondence:** Miriam Sinnhuber (miriam.sinnhuber@kit.edu)

**Abstract.**

The formation of NO by geomagnetic activity and EUV photoionization in the upper mesosphere and lower thermosphere and its subsequent impact on ozone contributes to the natural forcing of the climate system, and is recommended to be included in chemistry-climate model experiments since CMIP6. We compare NO concentrations simulated by five high-top chemistry-climate models – WACCM-X, EMAC, HAMMONIA, WACCM-D and KASIMA – in the mesosphere and thermosphere with satellite observations during a period of low geomagnetic and solar forcing from January to December 2010. While qualitatively the latitudinal and temporal variability of NO is captured by most models, we find disagreements of several orders of magnitude in high-latitude winter. Possible reasons are explored using snapshots at 12 UT on January 9, 2010. Two processes interacting with each other are identified as likely sources of these discrepancies, quenching of N($^2$D) by atomic oxygen in the mid-thermosphere, and meridional transport and mixing from the mid-thermosphere to the lower thermosphere. In the mid-thermosphere, the amount of atomic oxygen available from dissociation of molecular oxygen balances N($^4$S) and N($^2$D) via quenching of N($^2$D). N($^4$S) can then be transported or mixed into the lower thermosphere, where it efficiently reduces the lifetime of NO, leading to lower values of NO there. In high-latitude winter, meridional downward-poleward transport of N($^4$S) from the low-and midlatitude mid-thermosphere into the high-latitude lower thermosphere modulates the NO lifetime. This transport is affected by gravity waves, and therefore depends on the models gravity wave drag schemes and resolved gravity wave spectra.





## 1 Introduction

Precipitating energetic particles have been recognized as a source of nitric oxide in the high-latitude upper stratosphere, mesosphere and lower thermosphere since the $1960^{th}$ (e.g., Nicolet, 1965; Crutzen, 1975); recent reviews can be found in (Sinnhuber et al., 2012; Mironova et al., 2015; Baker et al., 2018). Similar processes also lead to the formation of NO in the low- and midlatitude uppermost mesosphere and lower thermosphere related to the absorption of solar electromagnetic radiation in the EUV and x-ray range (e.g., Watanabe et al., 1953; Barth, 1992; Marsh et al., 2004; Pettit et al., 2019). During polar winter, NO is long-lived and can be transported down from its source regions in the mesosphere and lower thermosphere into the upper stratosphere, contributing to ozone loss there (Funke et al., 2014; Randall et al., 2007; Sinnhuber et al., 2018). As ozone dominates radiative heating in the illuminated upper stratosphere and lower mesosphere and also contributes to radiative cooling, these changes in ozone initiate a chemical-radiative-dynamical coupling which even appears to affect large weather systems in high-latitude winter (Seppälä et al., 2009; Rozanov et al., 2012; Maliniemi et al., 2014, 2019). This so-called *indirect effect* of energetic particle precipitation (EPP) therefore contributes to the natural variability of the climate system, and consequently has been recommended to be included in climate model reconstructions and projections since CMIP6 (Matthes et al., 2017; Funke et al., 2024).

Starting point of the EPP indirect effect is the formation of nitric oxide mainly in the upper mesosphere and lower thermosphere by auroral and magnetospheric electron precipitation in high latitudes as well as by absorption of EUV and x-ray radiation. Dissociation and dissociative ionization of $N_2$ by collisions with energetic particles or absorption of EUV/x-ray radiation form atomic nitrogen in the ground ($N^4S$) or first excited ($N^2D$) state (see, e.g., Sinnhuber et al. (2012) and references therein[1]):

$$N_2 + hv, e^* \longrightarrow 2N(^2D), N(^4S) \qquad \text{(Eq. 1.1)}$$

$$N_2 + hv, e^* \longrightarrow N^+ + N(^2D), N(^4S). \qquad \text{(Eq. 1.2)}$$

Both the ground state $N(^4S)$ and the first excited state $N(^2D)$ of atomic nitrogen can react with molecular oxygen to form NO (Barth, 1992):

$$N(^4S) + O_2 \longrightarrow NO + O \qquad \text{(Eq. 2.1)}$$

$$N(^2D) + O_2 \longrightarrow NO + O \qquad \text{(Eq. 2.2)}$$

At temperatures below 400 K, reaction Eq. 2.2 is much faster than reaction Eq. 2.1, and NO is mainly formed via reaction Eq. 2.2. However, the rate constant of reaction Eq. 2.1 is strongly temperature dependent, and this reaction becomes a significant source of NO at temperatures above $\approx$ 400 K (see also discussion in Sinnhuber and Funke (2019)). Quenching of $N(^2D)$ by atomic oxygen or electrons has also been discussed:

$$N(^2D) + O \longrightarrow N(^4S) + O, \qquad \text{(Eq. 3.1)}$$

[1]Reactions Eq. 1.1 and Eq. 1.2 are discussed as *primary processes* in Sinnhuber et al. (2012) for energetic particles only, but are valid in the same way for EUV/X-ray radiation



$$N(^2D) + e^- \longrightarrow N(^4S) + e^-, \tag{Eq. 3.2}$$

and N($^2$D) also relaxes to N($^4$S) by fluorescence:

$$N(^2D) \longrightarrow N(^4S) + hv, \tag{Eq. 3.3}$$

see summaries and references in Barth (1992); Sinnhuber et al. (2012); Verronen et al. (2016).

Another source of NO is the formation of NO$^+$ by ion chemistry reactions summarized, e.g., in Barth (1992); Sinnhuber et al. (2012); Sinnhuber and Funke (2019):

$$N_2^+ + O \longrightarrow NO^+ + N(^2D), N(^4S) \tag{Eq. 4.1}$$

$$N^+ + O_2 \longrightarrow NO^+ + O \tag{Eq. 4.2}$$

$$O_2^+ + N_2 \longrightarrow NO^+ + NO \tag{Eq. 4.3}$$

$$O^+ + N_2 \longrightarrow NO^+ + N(^2D), N(^4S) \tag{Eq. 4.4}$$

followed by recombination again forming either N($^2$D) or N($^4$S).

$$NO^+ + e^- \longrightarrow N(^2D), N(^4S) + O. \tag{Eq. 5}$$

NO$^+$ can also be formed by photoionization of NO (Barth, 1992)

$$NO + hv \longrightarrow NO^+ + e^-. \tag{Eq. 6}$$

The main loss reactions for NO are the photolysis reaction

$$NO + hv \longrightarrow N(^4S) + O \tag{Eq. 7.1}$$

and the scavenging reaction with N($^4$S)

$$NO + N(^4S) \longrightarrow N_2 + O, \tag{Eq. 7.2}$$

see, e.g., (Barth, 1992; Marsh et al., 2004; Sinnhuber et al., 2012; Sinnhuber and Funke, 2019). The amount of NO formed due to particle or photo-ionization thus depends on the rate of ionization, but also on temperature and the partitioning between N($^2$D) and N($^4$S) formed – if the partitioning is in favour of N($^2$D), net NO formation is high, but if it is in favour of N($^4$S), enhanced loss due to reaction Eq. 7.2 could lead to a saturation effect with little net NO formation (Sinnhuber et al., 2012). Reactions Eq. 4.1 and Eq. 5 are expected to preferentially or solely produce N($^2$D), while reaction Eq. 4.4 produces mainly N($^4$S), and reactions Eq. 1.1 and Eq. 1.2 produce comparable amounts of N($^2$D) and N($^4$S) with partitionings between 0.4 and 0.6 (see, e.g., summaries and references in (Barth, 1992; Sinnhuber et al., 2012; Verronen et al., 2016)).

For chemistry-climate models with the top in the upper mesosphere, the EPP indirect effect is well described by an upper boundary condition prescribing either the flux of NO through the model top or the NO density at the model top, developed





by Funke et al. (2016) based on ten years of MIPAS observations; this is recommended for CMIP6 and CMIP7 (Matthes et al., 2017; Funke et al., 2024), and models using this upper boundary condition have been shown to reproduce NOy due to the EPP indirect effect very well (Sinnhuber et al., 2018; Arsenovic et al., 2019). High-top models with their top in the source region of auroral and EUV ionization which self-consistently consider NO formation by atmospheric ionization agree morphologically well, but mostly fail to reproduce the amount of NOy transported into the stratosphere (Smith-Johnsen et al., 2017; Funke et al., 2017; Sinnhuber et al., 2018; Pettit et al., 2019). Recently, a model-measurement intercomparison was carried out for a geomagnetic storm in April 2010 incorporating four high-top models extending into the lower thermosphere. This intercomparison has shown variations of up to one order of magnitude from model to model in the lower thermosphere even when using the same EUV and particle forcing (Sinnhuber et al., 2022). The overestimation of NO in the tropical lower thermosphere by three out of the four models compared to observations was tentatively interpreted as an overestimation of the rate of EUV photoionization provided by the parameterization of Solomon and Qian (2005) used in those models. A similar overestimation of low-latitude lower thermospheric NO has already been shown in a comparison of results of one model against observations of nitric oxide (Siskind et al., 2019), and was discussed as an indication of problems with the photochemistry there, as electron densities as another indicator of atmospheric ionization was underestimated by the model at the same time. The large spread between models in Sinnhuber et al. (2022) was tentatively interpreted as being due to differences in thermospheric temperature affecting the rate of formation of NO via Eq. 2.1. However, as the main focus of the Sinnhuber et al. (2022) intercomparison was on the impact of medium-energy electron precipitation onto mesospheric composition during the geomagnetic storm, this was not investigated further there.

Here, we follow up on the results of Sinnhuber et al. (2022) by carrying out a model intercomparison over a longer period of time to get a statistically more robust assessment of the models performance related to lower thermosphere NO, and by investigating in detail the roles of different reaction pathways forming and destroying NO using a snapshot of model results at one timestep.

## 2 Participating models and model experiments

### 2.1 Chemistry-climate Models

The same models participated in this follow-up experiment as in the Heppa III intercomparison discussed in Sinnhuber et al. (2022): WACCM-D, EMAC, HAMMONIA, and KASIMA. Additionally, results of WACCM-X are used here. WACCM-X shares the same chemistry code and derivation of ionization rates as WACCM-D, but has an extended model top and no detailed D-region ion chemistry. All participating models are *high-top* models with the model top well above the mesopause; all models use the same parameterization of EUV photoionization based on Solomon and Qian (2005), and most use particle impact ionization rates from the AISstorm model (see Sec. 2.2). Model tops vary from 115 km (KASIMA) to 600 km (WACCM-X), and the derivation of auroral ionization rates as well as the description of ion chemistry differ as well, see summary in Table 1 and detailed descriptions of all models below.





**Table 1.** Participating models. [1]: lower thermosphere ion chemistry with five positive ions and electrons. [2] depending on wavelength. [3] Verronen et al. (2016). [4]AISstorm 2.0: see Sec. 2.2. [5] assuming the partitioning of Porter et al. (1976) for photoionization and particles. [6]dissociation and dissociative ionization as described in Kieser (2011). [7] dissociation and dissociative ionization. [8]: assuming the partitioning of Jackman et al. (2005) for photoionization and particles, and assuming that the formation of NO equals the formation of N($^2$D).

| Model | Top [km] | Aurora | NO photo-ionization | Ion chemistry | N($^2$D)/N$_{tot}$ EUV | N($^2$D)/N$_{tot}$ particles |
|---|---|---|---|---|---|---|
| WACCM-X | 500 | internal | yes | LT[1] | 0.6/0.8[2] | 0.537[3] |
| EMAC | 200 | AISstorm 2.0[4] | no | LT[1] + O$_2^-$ | 0.485[5] | 0.485[5] |
| HAMMONIA | 180 | AISstorm 2.0[4] | yes | LT[1] | 0.6/0.5[6] | 0.6/0.5[6] |
| WACCM-D | 145 | internal | yes | D-region | 0.6/0.8[7] | 0.537[3] |
| KASIMA | 115 | AISstorm 2.0[4] | no | none | 0.56[8] | 0.56[8] |

**WACCM-D:** The Whole Atmosphere Community Climate Model Version 6 (WACCM6) is a chemistry-climate general circulation model that extends from the surface to about $6 \times 10^{-6}$ hPa (~140 km). The model horizontal resolution is 0.9° latitude by 1.25° longitude. A detailed description of the model physics in the MLT (mesosphere–lower thermosphere) region

is provided by Marsh et al. (2007). WACCM6 incorporates both the orographic and nonorographic (convective and frontal) gravity wave drag parametrisation (Richter et al., 2010). Here, we use WACCM6 in the specified dynamics configuration (FWmadSD) which is forced with meteorological fields (temperature and winds) from Modern-Era Retrospective analysis for Research and Applications (MERRA2, Molod et al. (2015)). Middle atmosphere D-region chemistry mechanism (MAD) is based on the Model for Ozone and Related Chemical Tracers, Version 3 (Kinnison et al., 2007). It represents chemical and

physical processes in the troposphere through the lower thermosphere. In addition to a six constituent ion chemistry model (O$^+$, O$_2^+$, N$^+$, N$_2^+$, NO$^+$, and electrons) that represents the E-region ionosphere, the MAD mechanism adds 15 positive and 21 negative ions with the aim to better reproduce the observed effects of energetic particle precipitation in the mesosphere and stratosphere (Verronen et al., 2016). For the solar spectral irradiance, geomagnetic indices, ion-pair production rates by galactic cosmic rays, solar protons, and medium-energy electrons, WACCM6 uses the recommended CMIP6 solar and geomagnetic

forcing as described in Matthes et al. (2017). For lower-energy electrons in the auroral regions, the model utilizes the auroral oval model by Roble and Ridley (1987). Photoionization and heating rates at wavelengths shorter than Lyman-$\alpha$ are based on the parameterization of Solomon and Qian (2005). Upper boundary conditions for temperature, H, O, O$_2$, N($^4$S) and N$_2$ are specified from the MSIS empirical model (Picone et al., 2002). NO at the upper boundary is specified from the Nitric Oxide Empirical Model NOEM (Marsh et al., 2004; Marsh et al., 2007).

**WACCM-X** is a superset of WACCM6 with its top boundary in the upper thermosphere ($4.5 \times 10^{-10}$ hPa, or ~600 km). It shares the same dynamics, physics and chemistry with WACCM6 up to the lower thermosphere, though the version of WACCM-X used in this study does not include D-region chemistry. At higher altitudes, the species-dependent dynamics,





thermospheric and ionospheric energetics, ionospheric electrodynamics and transport are included in WACCM-X (Liu et al., 2010, 2018, 2024b).

For the simulation used here, the high latitude electric potential and ion convection patterns are specified according to Heelis et al. (1982) parameterized by 3-hourly Kp input. No gravity wave parameterization is applied above ∼120 km, because the formulation based on linear saturation is no longer valid. Forcing data are applied in the same way as in WACCM-D with the exception of medium-energy electron ionization, which is included in the **Snapshot** model experiment, but not in the **Long** model experiment (see Sec. 2.3).

**EMAC:** The ECHAM/MESSy Atmospheric Chemistry model EMAC is an atmospheric composition-climate model which includes sub-models describing a wide range of atmospheric processes (Joeckel et al., 2010). EMAC uses the second version of the Modular Earth Submodel System (MESSy2) to link multi-institutional computer codes. The core atmospheric model is ECHAM5 (Roeckner et al., 2006). For the present study we used ECHAM5 version 5.3.02 and MESSy version 2.55.0 in the *upper atmosphere* mode with 74 vertical layers and a model top height of ≈220 km ($3e^{-7}$ hPa, EMAC submodule EDITH). The
horizontal resolution is T42, corresponding to a resolution of about $2.8° \times 2.8°$ in latitude and longitude. The model is nudged to the ECMWF ERA interim reanalysis data from the surface up to 1 hPa with decreasing nudging strength in a transition region in the six levels above. For orographic gravity waves, the parameterization of Lott and Miller (1997) is used. For non-orographic gravity waves, the Hines parameterization is used (Hines, 1997) in a set-up which allows propagation of gravity waves of ≈126 km horizontal wavelength and less than 12 km vertical wavelength into the lower thermosphere. Submodules
RAD and RAD-FUBRAD are used for radiative heating and cooling rates (Roeckner et al., 2003; Dietmüller et al., 2016), using the wavelength grid provided by FUBRAD for UV radiative heating in the upper mesosphere and thermosphere (Nissen et al., 2007; Kunze et al., 2014). For gas-phase reactions the submodule MECCA is used (Sander et al., 2011b, a), and photolysis rates are calculated with the JVAL submodule (Sander et al., 2014) which includes a parameterization for $O_2$ photodissociation in the Lyman-$\alpha$ range, but not in the Schumann-Runge bands and continuum. For NO photolysis, the parameterization from Allen
and Frederick (1982) is used without correction for self-absorption. For sensitivity studies, the $O_2$ photodissociation in the Schumann-Runge bands was implemented following Minschwaner et al. (1993), the $O_2$ photodissociation in the Schumann-Runge continuum was implemented with the same parameterization as used in KASIMA, but without consideration of the temperature dependence. Particle impact ionization rates for auroral electrons, auroral and solar protons and heavier ions are provided by 2-hourly results from the AISstorm 2.0 ionization model on the EMAC latitude/longitude and pressure grid.
EUV and x-ray photoionization rates are calculated based on the parameterization of Solomon and Qian (2005). A simple ion chemistry scheme is used to calculate the impact of particle impact and photoionization on the neutral composition, considering $O_2^+$, $N_2^+$, $O^+$, $N^+$, $NO^+$, electrons and $O_2^-$ as a placeholder for negative charge in the stratosphere and mesosphere.

**HAMMONIA:** the Hamburg Model of the Neutral and Ionized Atmosphere (HAMMONIA) is a chemistry-climate model that calculates interactions of atmospheric chemistry, radiation and dynamics from the surface to $3.4 \times 10^{-7}$ hPa (∼200-250
km). It consists of the ECHAM5 general circulation model (Roeckner et al., 2006) coupled to the MOZART3 chemistry module (Kinnison et al., 2007) and extended to the thermosphere (Schmidt et al., 2006; Meraner et al., 2016). HAMMONIA has 118 vertical levels and a T63 horizontal resolution, corresponding to about $1.9° \times 1.9°$ in latitude and longitude. For nudging, the





model uses ECMWF ERA interim reanalysis data from 850 hPa up to 1 hPa with an upper and lower transition zones. As in EMAC, for the orographic and and non-orographic gravity waves the model uses parameterizations of Lott and Miller (1997) and Hines (1997), respectively. Solar radiation is treated by a 6-band parameterization below 30 hPa (Cagnazzo et al., 2007) and by a 200-800 nm TUV parameterization (Madronich and Flocke, 1999) above, which is also used for photolysis calculations. In a 120-200 nm spectral region, the model uses various parameterizations for the $O_2$ photolysis including Schumann-Runge bands and continuum (for details, see Schmidt et al., 2006) and Minschwaner and Siskind (1993) for the NO photolysis. The ion chemistry consists of 13 ion-neutral reactions and 5 ion-electron recombinations involving $O_2^+$, $N_2^+$, $O^+$, $N^+$, $NO^+$, and electrons. This scheme is driven by the particle-induced ionization rates provided by the ionization model AISstorm 2.0 and by solar EUV and X-rays, following Solomon and Qian (2005). Joule heating and ion drag contribution to thermospheric temperature and wind tendencies are parameterized based on Zhu et al. (2005).

**KASIMA:** In this study we use the KArlsruhe SImulation Model of the middle Atmopshere (Kouker et al., 1999) in the version described in Sinnhuber et al. (2022). The model solves the meteorological basic equations in spectral form in the altitude range between 300 hPa and $3.6 \times 10^{-5}$ hPa with the pressure height $z = H \log(p/p_0)$ ($H = 7 \, \text{km}$ and $p_0 = 1013.25 \, \text{hPa}$) as vertical coordinate. It uses radiative forcing terms for UV-Vis and IR, and a gravity wave drag scheme. The model is relaxed (nudged) to ERA-Interim meteorological analyses (Dee et al., 2011) between the lower boundary of the model and 1 hPa. A full stratospheric chemistry including heterogeneous processes is adapted to include source terms related to particle and photo ionization. The ionization rates are taken from the AISstorm ionization model for the particle contribution, plus the photoionization based on the parameterization of Solomon and Qian (2005) which has been included in the model for this study. For the production of HOx the parameterization of Solomon et al. (1981) is used. For the production of NOx, 0.7 NO molecules are produced per ion pair and 0.55 N atoms in ground state.

## 2.2 Ionization model AISstorm 2.0

The Atmospheric Ionization during Substorms model AISstorm is a numerical model designed to calculate atmospheric ionization rates due to precipitating particles with high spatial resolution, improving upon its predecessor AIMOS (Wissing and Kallenrode, 2009) by specifically addressing substorm periods. AISstorm computes 3D ionization rates for precipitating protons, electrons, and alpha particles at a temporal resolution of 30 minutes. The model employs a sorting algorithm to allocate observations from polar-orbiting POES and Metop satellites into horizontal precipitation cells. To achieve this, AISstorm utilizes data from the TED and MEPED detectors and incorporates high-energy proton and alpha particle data from the SEM detectors on GOES satellites for the polar cap.

The energy range covered includes 154 eV to 500 MeV for protons, 154 eV to 300 keV for electrons, and 4 MeV to 500 MeV for alpha particles. Mean flux maps were generated from 18 years of satellite data (2001–2018), categorized by Kp level, geomagnetic APEX (Richmond, 1995), magnetic local time (MLT) location with up to 1° latitude by 3.75° longitude resolution, and substorm activity. Each flux map illustrates a typical spatial pattern of particle precipitation for a single particle channel on a global scale. Typical average flow maps are presented in Yakovchuk and Wissing (2019). The effective flow for a



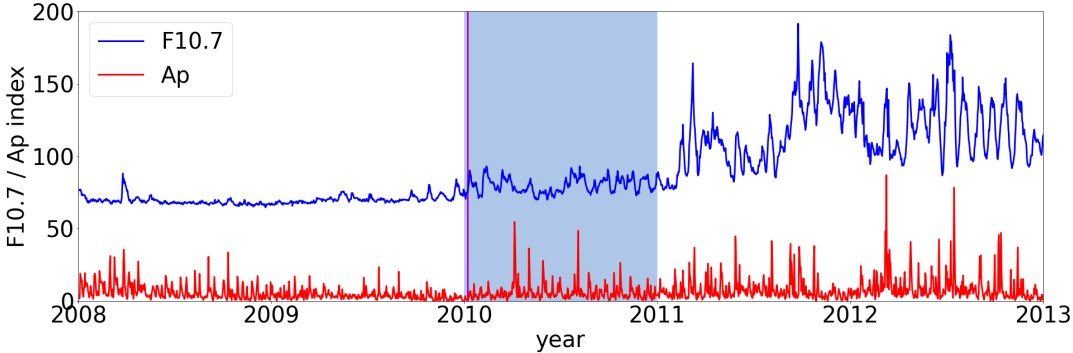

**Figure 1.** Daily F10.7 and Ap index for the period 2008–2013 from the CMIP6 forcing data-set. The blue box marks the period of the **Long** model run, the magenta line marks January 9, 2010, the date of the **Snapshot** model experiment.

30-minute interval is determined by scaling precipitation maps with direct measurements at that time, focusing on areas with high flux values (e.g., auroral oval) to minimize the impact of noise in real-time data.

For each interval, the ionization profile is calculated using the Monte Carlo method (Agostinelli et al., 2003; Schröter et al., 2006), with atmospheric parameters derived from the HAMMONIA (Schmidt et al., 2006) and NRLMSISE-00 (Picone et al., 2002) models.

## 2.3 Model experiments

Two main model experiments were set up and carried out by all models:

- For the **Long** experiment, model runs were carried out from January 1, 2009 to December 31, 2010. Model output was daily mean, zonal mean values of NO on the model pressure and latitude grids from January 1 to December 31, 2010, providing one year of data with a one-year spinup. The aim of this model experiment was to provide a statistically robust evaluation of the models performance in reproducing lower thermosphere NO compared to observations.

- The **Snapshot** model experiment branches off from the **Long** experiment, with output at 12:00 UT on January 9, 2010 on the models latitude, longitude and pressure grid. This allows a detailed analysis of the photochemical processes related to atmospheric ionization, in particular NO, N($^4$S), and electron density. January 9, 2010 was chosen as representing Northern hemisphere mid-winter covered by MIPAS UA observations.

An additional sensitivity experiment was carried out with EMAC only called **SRBC** in the following, repeating the spin-up and **Snapshot** experiment including a new representation of O$_2$ photodissociation in the Schumann-Runge bands and continuum as described in Section 2.1.

The year 2010 was chosen as an extension of the Heppa III period in April 2010 (Nesse et al., 2022; Sinnhuber et al., 2022). It is at the end of an extended solar minimum with very low solar and geomagnetic activity, see Fig. 1. Moderate geomagnetic



activity starts again in the second quarter of 2010 with auroral substorms and a moderate geomagnetic storm in April 2010, but EUV and x-ray fluxes remain low throughout the whole year. On the day of the **Snapshot** model run, EUV and auroral forcing are both relatively low.

## 3 Observational data-sets

To evaluate the models' performance in the lower thermosphere, model results are compared against satellite observations of parameters related to atmospheric ionization: nitric oxide NO and electron density.

### 3.1 Nitric oxide NO

Two data-sets of satellite observations of NO are used here which both scan in limb-observing mode into the lower thermosphere, MIPAS and SCIAMACHY, both on ENVISAT.

**MIPAS** measured thermal emission in the IR spectral range, scanning to 170 km in the UA/MA mode every 10 days. MIPAS observes independent of solar illumination on the day- and nightside of ENVISATs orbit with an equator crossing time of 10am/pm. We use the new calibration version 8, NO retrieval versions 561 and 662 (Funke et al., 2023). For comparison against the **Long** model run, daily average zonal averages in 10° bins are calculated as the mean of the am and pm daily mean values. For comparison against the **Snapshot** model experiment, daily zonal averages are calculated from the dayside (am) part
of the orbits only.

**SCIAMACHY** observed resonance fluorescence of NO in the $\gamma$ band emissions, scanning up to 150 km every 15 days (MLT mode), overlapping with MIPAS MA/UA observations once per month (Bender et al., 2013, 2015). Data are gridded along the orbit, and daily zonal means have been calculated for daytime (sza≤88°) with averaging kernel diagonal element ≥0.02. As SCIAMACHY observations depend on solar light, no night-time data or data in polar night are available. SCIAMACHY data
are therefore used only for comparison against the **Long** model experiment.

### 3.2 Electron density

Electron densities are provided from radio occultation observations by FORMOSAT-3/COSMIC–1, which are freely available at https://data.cosmic.ucar.edu/gnss-ro/cosmic1/repro2021/level2/. The orbit of the FORMOSAT-3 satellites has an inclination of 72°, so observations at high latitudes are sparse compared to mid- and low-latitudes. In the altitude range 90–120 km,
sporadic E-layers frequently occur particularly during local afternoon in mid-latitude summer. These result in enhancements of the electron density in sharp, distinctive layers (e.g. Arras et al., 2022). In low latitudes, ionospheric scintillations are associated with a strong variation of the electron density leading to very low or even negative values in the observed density profile (e.g. Kepkar et al., 2020). The scintillations are frequently caused by equatorial plasma depletions in the F–region at altitudes between 250–500 km that predominantly occur after sunset. Both processes are not considered in the ion chemistry
schemes of the models used here. To emulate the model output on noon of January 9, 2010, all observations of January 9, 2010, were therefore screened in the following way. In a first step, only observations with local solar times between 9-15 hours were





selected. All individual profiles with values ≤ zero or NaNs between 92 km and 205 km were rejected, as were all profiles with vertical gradients from one vertical layer to the next of more than 35 % of a running mean over 7 vertical layers. In this way, smooth daily average profiles around local noon are provided (see upper left panel of Fig. 3. Note the limited coverage of high

latitudes as well as a data gap in the Northern subtropics, which is due to the local time sampling.

## 4  Results

### 4.1  Assessment of modelled NO, January-December 2010

In Figure 2, model results of NO from all five models are shown compared against NO observations from MIPAS and SCIA-MACHY for the year 2010 in two latitude bins: in the tropics (0-10°N), and in high Northern latitudes (70-80°N). MIPAS

data are means of am (dayside) and pm (nightside) measurements. SCIAMACHY data are am (dayside) only. Model results are averaged over the whole day. A comparison of MIPAS am and pm data shows differences generally within a factor of two, with sporadically larger differences up to a factor of 10 presumably related to differences in different sampling on the dayside and nightside of the orbit (not shown); systematic order-of-magnitude differences due to the difference in daily averaging are therefore not expected.

In the low latitude lower thermosphere, NO is formed mainly by photoionization in the EUV and x-ray spectral range. Both observational data-sets show a distinct layer of NO between $10^{-2}$ and $10^{-5}$ hPa, with largest values of $(2.5–7.5) \times 10^7 \mathrm{cm}^{-3}$. The temporal coverage is different between MIPAS and SCIAMACHY, and SCIAMACHY data are daytime only; also SCIA-MACHY data appear to be more noisy and show more variability in particular in the vertical structure. Despite this, both observational data-sets agree both quantitatively and morphologically very well. The models also all show clear NO layers in

the lower tropical thermosphere with little temporal variation. However, the size, position and strength of the NO layer is different from the observations. In WACCM-X, the absolute numbers of the NO layer are captured quite well, being in the range of $(2.5-5) \times 10^7 \mathrm{cm}^{-3}$. However, the NO layer is more narrow in altitude, clearly confined to $10^{-3}$-$10^{-4}$ hPa, so the overall amount of NO is probably lower than in the observations. Results from all other models show significantly higher NO values than the observations, with highest values of $(1-2.5) \times 10^8 \mathrm{cm}^{-3}$ in HAMMONIA and WACCM-D. EMAC and KASIMA reach

maximal values of $(7.5-10) \times 10^7 \mathrm{cm}^{-3}$. In EMAC and HAMMONIA, the NO maximum is broader than in the observations, reaching further up into the lower thermosphere.

At high latitudes, observations also show a distinctive NO layer in the lower thermosphere with higher maximal values, up to $(1-2.5) \times 10^8 \mathrm{cm}^{-3}$ during spring to autumn, up to $(2.5-5) \times 10^8 \mathrm{cm}^{-3}$ during November and December 2010 (only covered by MIPAS). Enhanced values of NO of up to $(2.5-5) \times 10^8 \mathrm{cm}^{-3}$ (MIPAS) respectively up to $(7.5-10) \times 10^8 \mathrm{cm}^{-3}$ (SCIAMACHY)

are observed in the upper mesosphere in February and March 2010, indicating downward coupling via transport or mixing across the polar winter mesopause. The models qualitatively show a similar behaviour, with higher values in the lower thermo-spheric NO layer at high latitudes than in the tropics, and downward coupling into the upper mesosphere in February 2010 there. WACCM-X clearly underestimates NO in the high latitude lower thermosphere particularly during polar winter, with values falling below $1 \times 10^6 \mathrm{cm}^{-3}$ in early January, for a short period even below $1 \times 10^5 \mathrm{cm}^{-3}$. All other models show too high values



**Figure 2.** Zonal mean daily mean NO in the lower thermosphere throughout the year 2010. Left: tropics (0-10°N). Right: high Northern latitudes (70-80°N). From top to bottom: MIPAS observations, SCIAMACHY observations, model results from WACCM-X, EMAC, HAM-MONIA, WACCM-D and KASIMA. SCIAMACHY data are 10 am local solar time in the illuminated part of the orbit, MIPAS is the mean of 10am/pm observations, and the model results are true daily averages.





in the thermospheric NO layer, with highest values of up to $(5\text{-}7.5)\times10^8\text{cm}^{-3}$ reached in EMAC. WACCM-D shows good agreement during polar winter, but too high values compared to observations during the summer season; KASIMA shows good agreement during summer, but too high values during winter. HAMMONIA generally agrees well apart from short episodes of higher NO during one to two days which might be due in part to the low temporal sampling of the observations. EMAC and WACCM-D show relatively constant values over the summer period, while HAMMONIA, KASIMA and WACCM-X show a

higher amount of day-to-day variability which is more consistent with the observed variability. The mesospheric intrusion of NO during March 2010 is overestimated by most models, with highest values shown by EMAC, good agreement with MIPAS observations shown by WACCM-D and HAMMONIA, and too low values shown by WACCM-X.

In summary, all models reproduce the main features of the thermospheric NO variability, showing a distinct thermospheric NO layer roughly in the correct pressure region, with higher values at high latitudes than at low latitudes, and with an intrusion

from the thermospheric NO layer into the upper mesosphere during polar spring. However, all models fail to reproduce the observations in the upper mesosphere and lower thermosphere quantitatively. The best quantitative agreement in low latitudes and during polar summer is provided by WACCM-X, which however underestimates NO during high-latitude winter by orders of magnitude. All other models show too high values of NO in the lower thermospheric NO layer, leading to an overestimation of the mesospheric intrusion during late winter. The qualitative and quantitative difference between WACCM-X and all other

models is particularly noteworthy as WACCM-X uses the same parameterizations for auroral and EUV photoionization and the same photochemistry scheme as WACCM-D. This suggests that the source of the large discrepancies in lower thermospheric NO between WACCM-X and WACCM-D (and by inference, also to the other models) lies in the mid-thermosphere, above the top altitude of WACCM-D.

This is investigated in more detail in the following Section 4.2.

**4.2 Ionization, NO photochemistry, and thermospheric dynamics**

In this section, the **Snapshot** model experiment is analysed in detail to determine the differences in NO formation and loss related to lower thermospheric ionization. In a first step, electron densities are compared against observations to assess the validity of the ionization rate forcing (Section 4.2.1). In a second step, NO is compared against observations (Section 4.2.2), the mechanism of N and NO formation and loss and their differences between the different models are investigated (Section 4.2.3),

and finally, the role of thermospheric dynamics is discussed with a focus on the winter hemisphere mid-to high latitudes (Section 4.2.4).

**4.2.1 Electron densities as a measure of ionization rate**

In the upper left panel of Figure 3, electron densities in the lower thermosphere from COSMIC-1 are shown as a latitude/altitude cross-section for January 9, 2010. Due to the orbit of the FORMOSAT-3 satellites with an inclination of 72°, coverage of the

auroral regions is limited to the outer edge of the Southern auroral oval. Observational gaps in Northern low latitudes are due to the local time sampling. Not considering sporadic E-layers and ionospheric scintillations, the observed distribution of electron density in the lower and mid-thermosphere between 90-200 km is fairly regular, with a steep increase in altitude below, but a





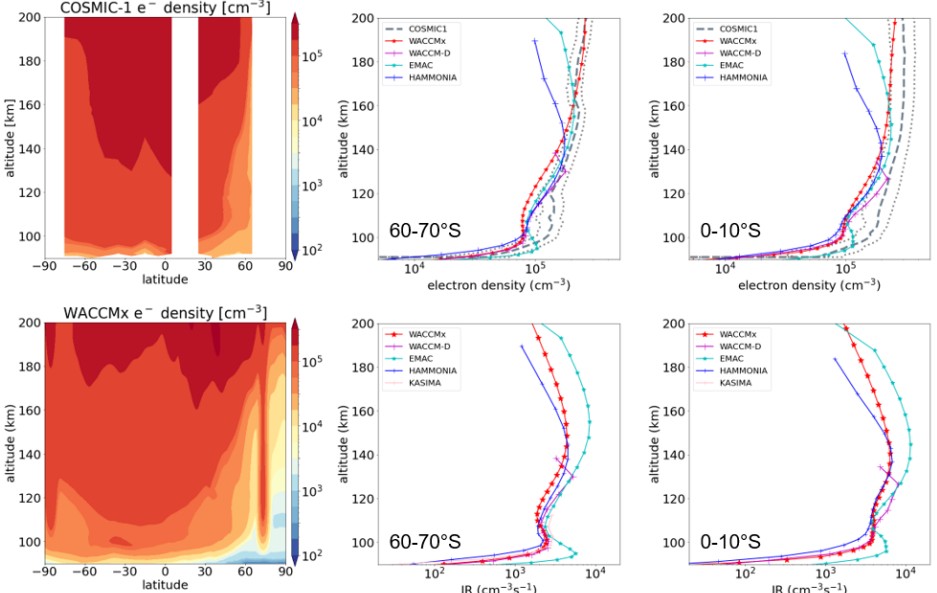

**Figure 3.** Upper panels, left: daily mean (9-15 LST) zonal mean electron densities from COSMIC-1 on Januar 9, 2010, after correction for sporadic-E and scintillations. Upper middle and right panels: comparison of COSMIC-1 electron densities zonally averaged for local times 9-15 hours at 60-70°S (middle) and 0-10°S (right) with model results of the **Snapshot** model experiment at 12:00 UTC, 0°E, averaged over the same 10° wide latitude bins. The light gray lines denote the COSMIC-1 3 $\sigma$ standard error of the mean. Lower panels, left: WACCM-X electron density at 12:00 UTC, along 0°E on January 9, 2010. Lower panels, middle and right: total (particle and photo-) ionization rates of the models at 12:00 UTC, 0°E, for 0-10°N and 60-70°S.

slow increase in altitude above 100 km, a decrease of values into polar night in high Northern latitudes, and maximal values in 60-0°S in 140-180 km altitude. All models qualitatively reproduce this behaviour; a latitude/altitude cross section of the same

320 day is shown at 12 UT along the 0° meridian exemplarily for WACCM-X. For a quantitative comparison, model results from all models but KASIMA, which does not explicitly consider ion chemistry, are shown at high and low Southern latitudes (0-10°S and 60-70°S), compared with COSMIC-1 data averaged over the same latitude regions. The vertical structure of the electron density is reproduced qualitatively well by all models. However, in 100-120 km, all models underestimate electron densities in both latitude bins by 10-50%. In 120-140/160 km altitude, WACCM-D, HAMMONIA (140 km) and EMAC (160 km) are

325 within the error range of the observations, though lower than the mean value, while WACCM-X in this altitude range has lower values, just outside the error range of the observations. Above these altitudes, HAMMONIA and EMAC show significantly lower values than the observations, while WACCM-X is in very good agreement.

 Electron densities and ionization rates in the lower thermosphere are closely correlated (compare lower middle and right panels to upper middle and right panels of Figure 3), forming a compact log-log distribution (not shown). This suggests that

330 electron densities can be used as an indicator of the rate of ionization in the lower thermosphere. In this sense, in 100-120 km





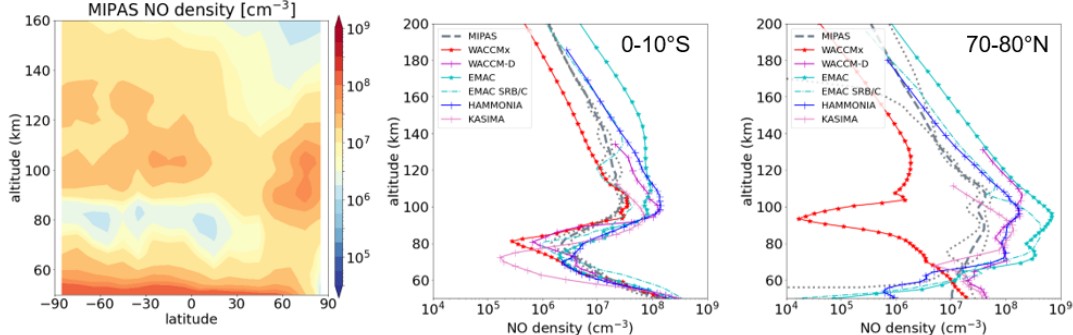

**Figure 4.** Left: MIPAS zonal mean daily mean daytime NO on January 9, 2010 in the upper mesosphere and thermosphere. Middle and right: MIPAS NO compared against model results from the **Snapshot** model experiment in 0-10°S and 70-80°N. The error range is the $3\sigma$ standard error of the mean. Also shown are results of the **SRB/C** model experiments of EMAC.

altitude, all models likely underestimate the rate of ionization, but are roughly in agreement above, with a better agreement of WACCM-D, HAMMONIA and EMAC in 120-140/160 km, a better agreement of WACCM-X above these altitudes. In the latitude ranges shown here, ionization is mainly due to EUV photoionization, and the underestimation of the electron densities in 100-120 km altitude, as well as the distinct peaks in atmospheric ionization below this altitude in 90-100 km in all models, might indicate a systematic problem either of the EUV photoionization parameterization, or of the radiative transfer, in all models. However, as all models are in agreement to, or lower than, the electron density observations, the ionization rates are likely not the reason for the overestimation of nitric oxide in the lower thermosphere by WACCM-D, HAMMONIA and EMAC shown in the previous Section 4.1.

### 4.2.2 Nitric oxide

In Figure 4, NO densities of MIPAS daytime observations are shown for January 9, 2010, compared to model results in two latitude bins, in low Southern and high Northern latitudes.

In low latitudes, observations show a sharp increase of NO into the lower thermosphere with maximal values around 100 km, and a slow decrease with altitude above. All models reproduce the morphology well, but fail to reproduce absolute values; WACCM-X is in good agreement with observations around 100 km altitude but has significantly lower values above, while all other models overestimate NO compared to observations above 90 km altitude, with highest values below 120 km by HAMMONIA, above 120 km by EMAC.

At high Northern latitudes, NO shows a broader maximum extending down into the upper mesosphere, indicative of thermosphere-mesosphere coupling in polar winter, and values decreasing with altitude above 110 km. KASIMA, WACCM-D, HAMMONIA, and EMAC qualitatively reproduce this, but show significantly higher values, with highest values shown by EMAC. WACCM-X shows a decrease with altitude from the mesosphere into the lower thermosphere, with a distinct mini-



mum around 90-100 km and a steep increase above; however, values remain lower than the observations or the other models by about one order of magnitude throughout the whole altitude range.

This is consistent with results of the **Long** model runs shown in Section 4.1. Considering the intercomparison of electron densities shown in the previous Section 4.2.1, the differences in NO between models and observations on the one hand,
WACCM-X and all other models on the other hand, can not be explained by differences in the ionization forcing. Differences in either the photochemistry of $N(^4S)$ and NO above the top of WACCM-D or thermospheric dynamics are likely reasons. These are investigated in the following two sections.

### 4.2.3 Photochemical formation and loss of $N(^4S)$, $N(^2D)$, and NO

In Figure 5, NO, $N(^4S)$, $N(^2D)$ and the photochemical lifetime of NO are shown for all models along the $0°$ meridian at 12 UTC
on January 9, 2010. The comparison of NO highlights again the features already discussed in previous sections: lowest values in WACCM-X with a distinct minimum in the Northern high-latitude lower thermosphere and upper mesosphere, higher values in all other models with a distinct maximum in the polar winter high latitudes extending well into the mesosphere, which is particularly pronounced in EMAC. $N(^4S)$ and $N(^2D)$ show a sharp increase at the mesopause in all models, with values increasing with altitude within the lower thermosphere. Values of $N(^2D)$ are in the same order of magnitude in their common
altitude ranges, indicating that ionization rates and reactions forming $N(^2D)$ are not substantially different. $N(^4S)$ shows a maximum in the mid-thermosphere (140-160 km in WACCM-X and HAMMONIA, above 160 km in EMAC) in the three models extending above 150 km. Up to 140 km, values of $N(^4S)$ are similar in KASIMA, WACCM-D, HAMMONIA, and EMAC, while WACCM-X shows significantly higher values. The difference in the amount of $N(^4S)$ between WACCM-X and the other models has implications also for the photochemical lifetime of NO, as the reaction of $N(^4S)$ with NO (reaction Eq. 7.2)
is the main loss process of NO. Lifetimes of NO considering reaction Eq. 7.2, NO photodissociation and NO photoionization are shown in the right-hand panels of Figure 5 and show significantly lower NO lifetimes for WACCM-X in the lower to mid-thermosphere at all latitudes, and in the high-latitude polar winter lower thermosphere, clearly anti-correlated with higher values of $N(^4S)$. Lower values of NO in WACCM-X in the illuminated mid-thermosphere above 140 km as well as in the polar winter lower thermosphere compared to the other models can therefore be explained by larger abundances of $N(^4S)$ in these
altitudes. The two models with their tops below the mid-thermosphere $N(^4S)$ maximum have upper boundary conditions of NO and N; for those models, increased $N(^4S)$ at the upper boundary could probably improve the representation of NO in the low-to midlatitude lower thermosphere. However, this can not explain the discrepancy between WACCM-X on the one hand, HAMMONIA and EMAC on the other hand, which have their model tops in or above this maximum.

In Fig. 6, the rates of the reactions of $N(^4S)$ with $O_2$ forming NO (the rate constant of reaction Eq. 2.1 times the $O_2$ density),
$N(^2D)$ with $O_2$ forming NO (rate constant of reaction Eq. 2.2 times $O_2$ density), and $N(^2D)$ with O forming $N(^4S)$ (rate constant of reaction Eq. 3.1 times O density) are shown, calculated from the results of the **Snapshot** model experiments of NO, $N(^4S)$, $N(^2D)$, O, $O_2$ and temperature at 12:00 UTC on January 9, 2010, along the $0°$ meridian. Only those models with their top above 150 km are shown here. For WACCM-X, the rates of all three reactions fall in a similar range of values, with maximal values of (4000-8000) $cm^{-3}s^{-1}$ around 120-160 km. In EMAC and HAMMONIA, the rate of the reaction $N(^2D)$ with $O_2$ is distinctly



**Figure 5.** Snapshots of NO (left), N($^4$S), N($^2$D), and the photochemical lifetime of NO from the **Snapshot** model experiment on January 9, 2010, 12:00 UTC, at 0°E. From top to bottom: WACCM-X, HAMMONIA, WACCM-D and KASIMA. Note KASIMA does not consider N($^2$D) explicitly, but instead assumes that all N($^2$D) immediately form NO.



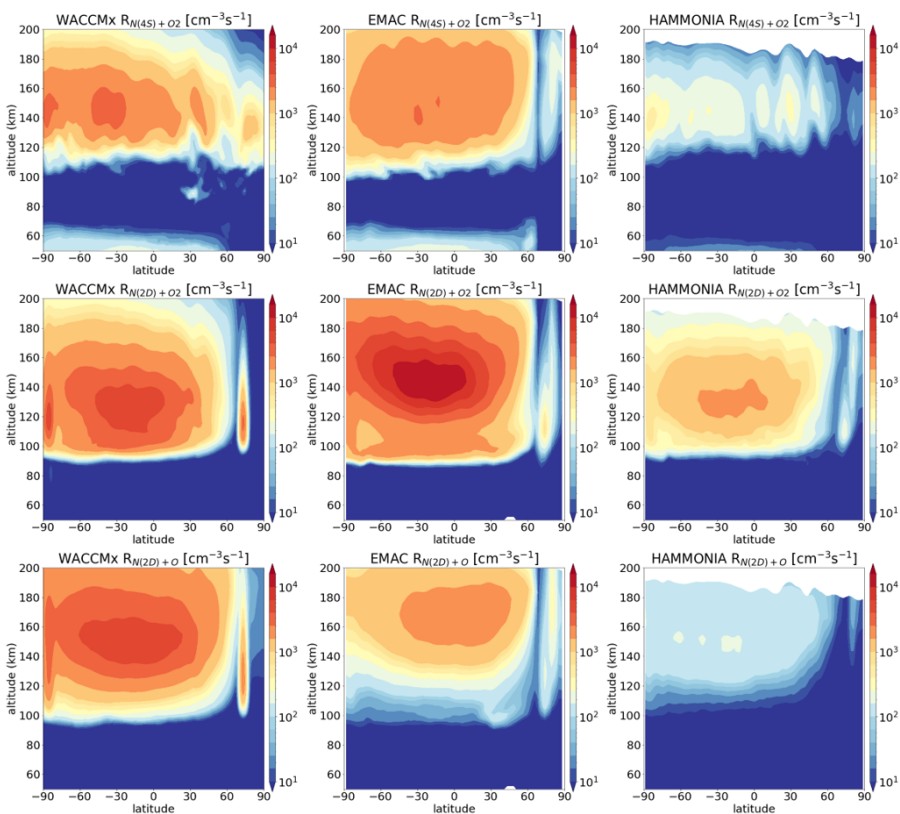

**Figure 6.** Rates of the reactions of (from left to right) N($^4$S) and O$_2$ (Eq. Eq. 2.1), N($^2$D) and O$_2$ (Eq. Eq. 2.2), and N($^2$D) and O. From left to right: WACCM-X, EMAC, and HAMMONIA. **Snapshot** model experiment on January 9, 2010, 12:00 UTC and 0°E.

faster than the rates of the other two reactions. In HAMMONIA, all rates are distinctly slower than in WACCM-X or EMAC.
In EMAC, the rate of the reaction of N($^2$D) with O$_2$ is also significantly faster than the rate of the same reaction in WACCM-X,
while the rate of the reaction N($^2$D) with O is significantly slower than the rate of the respective reaction in WACCM-X. As the
amount of N($^2$D) is comparable between the two models in the respective altitude ranges, this indicates a significantly different
ratio of atomic oxygen to molecular oxygen. The ratio of O to O$_2$ is shown for WACCM-X, HAMMONIA and EMAC in

the upper panels of Figure 7, confirming that this ratio is much lower in EMAC and HAMMONIA than in WACCM-X. In
WACCM-X, the unity line is in the lowermost thermosphere around 100 km in all latitudes, while in EMAC, it ranges from
above 190 km in the high-latitude Southern hemisphere to around 110 km in the high-latitude Northern hemisphere, and in
HAMMONIA, it is between 135–160 km, with little horizontal variation. Atomic oxygen in the thermosphere is produced by
photodissociation of O$_2$ in the Schumann-Runge bands, Schumann-Runge continuum, and Lyman-$\alpha$ range as well as by EUV

photodissocation of O$_2$. The rate of EUV photodissociation in all models is based on Solomon and Qian (2005), and therefore
should not differ significantly. However, EMAC does not consider photodissociation of O$_2$ in the Schumann-Runge bands



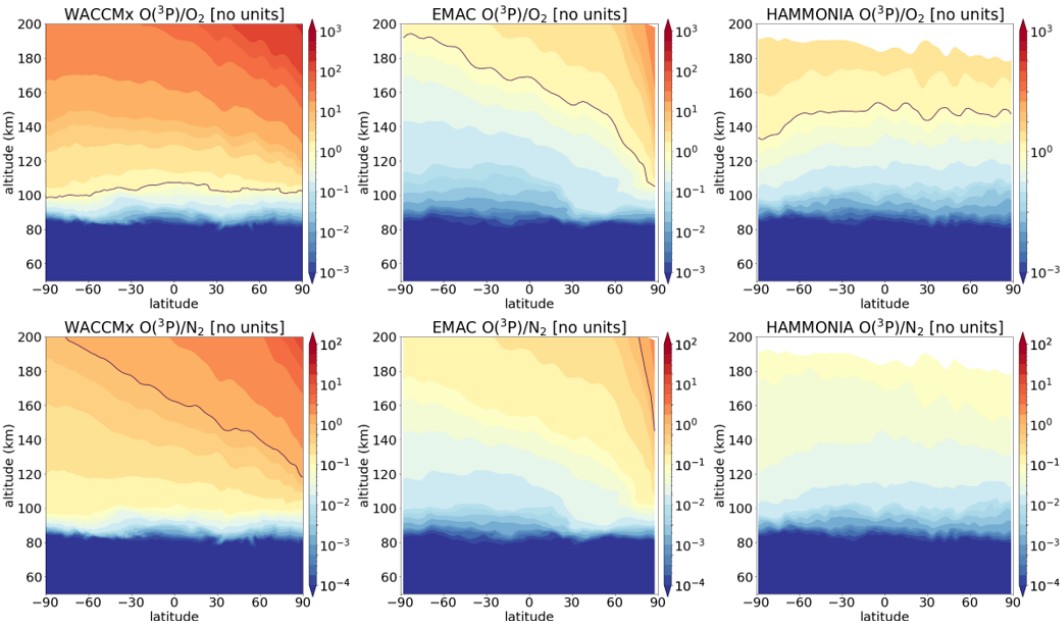

**Figure 7.** Ratio of O to $O_2$ (top) and O to $N_2$ (bottom) for (from left to right) WACCM-X, EMAC, and HAMMONIA. **Snapshot** model experiment on January 9, 2010, 12:00 UTC and 0°E.

and continuum, while this is included, e.g., in WACCM-X and HAMMONIA. The difference in the O to $O_2$ ratio between WACCM-X and EMAC can therefore presumably be explained by missing photodissociation of $O_2$ in the Schumann-Runge bands and continuum in EMAC. As the ratio between O and $O_2$ determines the balance between formation of NO or $N(^4S)$ by

$N(^2D)$, this is then also the source of the discrepancy in $N(^4S)$ between the two models; the amount of $N(^4S)$ in turn determines the amount of NO due to its impact on the lifetime of NO.

To test this, an additional model experiment was carried out with EMAC including simple parametrizations of $O_2$ photodissociation in the Schumann-Runge bands and continuum (experiment **SRBC**). Results from this experiment for NO, $N(^4S)$ and the ratio of O to $O_2$ are shown compared to the **Snapshot** experiments for 12:00 UTC on January 9, 2010 along the 0°

meridian in Figure 8. It is shown that NO in the thermospheric NO layer decreases significantly when increasing the rate of $O_2$ photodissociation (compare, e.g., to Figure 5); when Schumann-Runge bands and continuum are considered, NO in the lower thermosphere is in much better agreement with observations as well as with results from WACCM-X in the Southern (summer) hemisphere and in low- and mid-latitudes of the Northern (winter) hemisphere, see also Figure 4. $N(^4S)$ and the ratio of O to $O_2$ increase, and are in much better agreement with WACCM-X values for the **SRBC** case; the unity line of O to $O_2$ is now

around 120 km altitude. However, significantly too high values of NO compared to observations persist in EMAC in the polar winter lower thermosphere and upper mesosphere.



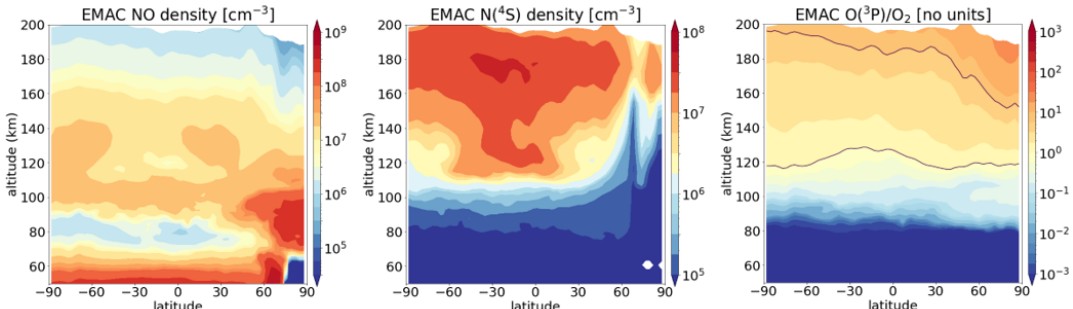

**Figure 8.** Snapshots of NO (left), N($^4$S) (middle) and the ratio of O to O$_2$ (right) for the EMAC model experiment **SRB/C** at 12:00 UTC on January 9, 2010, along the 0° meridian. Comparison to Figures 5 and 7 show a generally better quantitative agreement with WACCM-X, though differences persist particularly in high Northern latitudes.

#### 4.2.4 Lower thermosphere dynamics and the polar winter lower thermosphere

Atomic oxygen is produced by photodissociation and photoionization of O$_2$ in the lower thermosphere, and the ratio of O to O$_2$ increases with increasing altitude, reflecting increasing transition of O$_2$ to O. As this transition depends on solar illumination, highest values would be expected in the region of strongest illumination: in polar summer and tropical regions. However, this is not the case in WACCM-X and EMAC - in the mid-thermosphere above 150 km, both show an increase in values of the O to O$_2$ ratio into polar night (upper panel of Fig. 7). This suggests downward and poleward transport and mixing from the mid-latitude mid-thermosphere at 140 km to 200 km to the high-latitude lower thermosphere below 140 km.

More commonly, the ratio of O to N$_2$ is used as an indicator of vertical motions in the lower thermosphere (e.g., Fuller-Rowell, 1998). One advantage of using the O to N$_2$ ratio is that observations of the thermospheric column of this ratio are available for model evaluation, e.g., from GUVI/TIMED (https://guvitimed.jhuapl.edu/). However, those observations do not cover high Northern latitudes on January 9, 2010, so can not be used here. Another aspect to note is that the O to N$_2$ ratio is affected not only by the rate of photodissociation of O$_2$ forming O, but also by the treatment of N$_2$, which is very different in the three models: WACCM-X and HAMMONIA treat N$_2$ as a fill gas, EMAC treats N$_2$ as a full chemical tracer. Molecular diffusion leads to a distinct N$_2$ layer in the lower thermosphere in EMAC, presumably contributing to the lower O to N$_2$ ratio compared to WACCM-X. The O to N$_2$ ratio is shown for WACCM-X, EMAC, and HAMMONIA in the lower panels of Figure 7, and shows a mainly consistent behaviour to the O to O$_2$ ratio.

Very different scenarios for the meridional motions in the lower to mid thermosphere between the three models are indicated by the O to O$_2$ and O to N$_2$ ratios. For WACCM-X, gradually descending contour lines from the tropical mid-thermosphere to the polar winter lower thermosphere indicate gradual continuous transport and mixing from the tropical mid-thermosphere to the polar winter lower thermosphere, which efficiently transports N($^4$S) from its main source region in the tropical mid-thermosphere into the polar winter lower thermosphere. The very low values of NO shown in WACCM-X in the polar winter



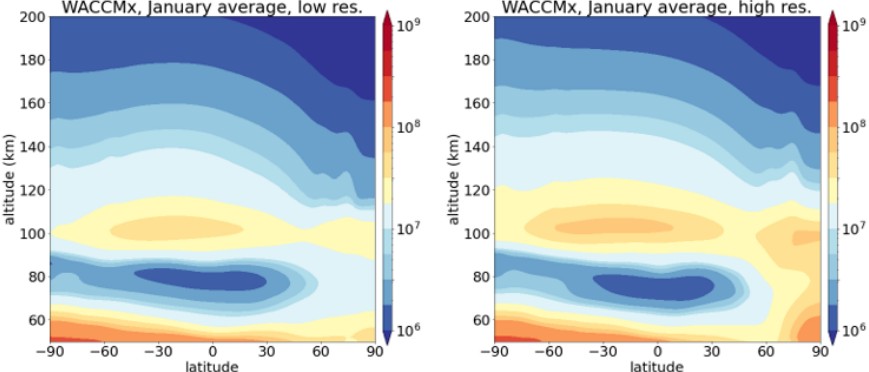

**Figure 9.** Monthly mean zonal mean January values of NO from two free-running WACCM-X model experiments with moderate (≈200 km, left) and high (≈25 km, right) resolution under constant moderate solar conditions. The model experiments are described in Liu et al. (2024a). The comparison of polar winter mesospheric and thermospheric NO highlights the impact of model resolution and resolved gravity waves on NO in the lower thermosphere and high-latitude winter lower thermosphere and upper mesosphere.

lower thermosphere are therefore likely a combination of strong formation of N($^4$S) from N($^2$D) quenching with O in the tropical and subtropical mid-thermosphere, and downward and poleward transport of N($^4$S) from the source regions to the
winter hemisphere lower thermosphere. In EMAC, contour lines of O to $N_2$ (O to $O_2$) over the winter pole are much steeper than in WACCM-X, and there is a change in the poleward/downward gradient around 60°N. This indicates downward transport mainly over the winter pole, effectively suppressing transport of N($^4$S) from the source region in the mid-and low latitude mid-thermosphere into the polar winter lower thermosphere. Note this change in gradient at the edge of the polar night terminator persists also in the **SRBC** experiments, and a lack of poleward/downward transport or mixing can explain the persisting high
values of NO in the polar winter lower thermosphere in these experiments. In HAMMONIA, the ratio of O to $N_2$ (O to $O_2$) does not indicate strong downward or poleward transport or mixing in the winter thermosphere.

Comparison with NO observations, as discussed in previous sections, indicate that the amount of N($^4$S) in the winter polar lower thermosphere is likely too high in WACCM-X, too low in EMAC. This suggests that at least some poleward/downward transport and mixing occurs, which however is overestimated in WACCM-X, underestimated in EMAC and HAMMONIA.
Liu et al. (2024a) discuss a possible impact of gravity wave drag in the thermosphere on thermospheric circulation in both the summer and winter hemisphere. They have shown that the thermospheric circulation is better reproduced in WACCM-X in a setup with higher spatial resolution, leading, e.g., to a better representation of the column O to $N_2$ ratio presumably because in this setup, a larger part of the gravity wave spectrum is resolved including secondary and tertiary gravity waves forming in the thermosphere (Becker and Vadas, 2020) which are not captured by gravity wave parameterizations. The more realistic
representation of thermospheric transport also leads to a better representation of NO particularly in the polar winter lower thermosphere, see Figure 9. The gravity wave parameterization in WACCM-X prevents the propagation of parameterized gravity waves beyond 120 km, while in EMAC and HAMMONIA, the gravity wave drag is greatly reduced in the thermo-

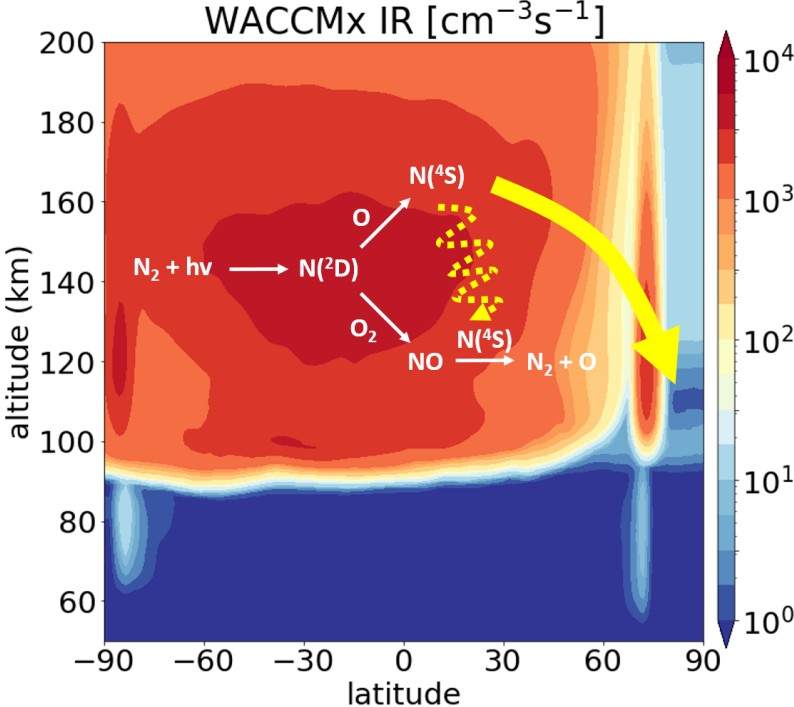

**Figure 10.** Schematic view of the processes important for NO formation and loss during solar minimum conditions. Dissociation of $N_2$ by EUV – at high latitudes also energetic particles – leads to the formation of N in the excited states. In the lower thermosphere, $N(^2D)$ preferentially reacts with $O_2$ forming NO, but in the mid-thermosphere, reaction with O dominates forming $N(^4S)$. In mid- and low latitudes, $N(^4S)$ is mixed down into the thermospheric NO layer by molecular diffusion (dotted yellow line). In the winter hemisphere, it can also be transported downward and poleward (thick yellow arrow) in a meridional circulation presumably limited by secondary and tertiary gravity waves. Finally, NO is destroyed by reaction with $N(^4S)$, so the transport and mixing of $N(^4S)$ from the mid-thermosphere modulates the amount of NO in the lower thermosphere. The underlying figure is the total rate of ionization considering EUV photoionization and particle impact ionization from WACCM-X on January 9, 2010, at 12:00 UT along the $0°$ meridian.

sphere compared to the mesopause region, but is not totally supressed. The inference is that gravity wave drag decelerates the thermospheric meridional winter circulation. However, validating this statement is beyond the scope of this paper, and the thermospheric circulation and its impact on lower thermospheric NO and the EPP indirect effect should be investigated in more detail in the future.

## 5  Summary and conclusions

Consistent with results of Sinnhuber et al. (2022), we show significant differences in lower thermospheric NO between different chemistry-climate models as well as in comparison to satellite observations. In the low-latitude lower thermosphere, differences



are in the range of one order of magnitude, with KASIMA, WACCM-D, HAMMONIA and EMAC showing higher values than observations, while WACCM-X is in range of, or lower than, the observations. In the polar winter lower thermosphere and upper mesosphere, differences reach four to five orders of magnitude between WACCM-X on the one hand, EMAC, HAMMONIA, WACCM-D and KASIMA on the other hand, with highest values shown by EMAC, and the MIPAS observations being lower than KASIMA, WACCM-D, HAMMONIA, and EMAC, but significantly higher than WACCM-X. Comparison of electron

densities as an indicator of atmospheric ionization show that these differences can not be explained by differences in the ionization forcing.

Apart from atmospheric ionization, two processes control the amount of NO in the lower thermosphere: The formation of $N(^4S)$ by photodissociation of $N_2$ in the illuminated mid-thermosphere, and downward transport and mixing of $N(^4S)$ into the NO layer. EUV photodissociation of $N_2$ produces atomic nitrogen in the ground ($N(^4S)$)) and excited ($N(^2D)$)) state. In the

lower thermosphere, $N(^2D)$ reacts with $O_2$ forming NO very efficiently (reaction Eq. 2.2). In the mid-thermosphere, where atomic oxygen is more abundant than molecular oxygen, the competing reaction of $N(^2D)$ with O forming $N(^4S)$ becomes comparatively more important, leading to formation of $N(^4S)$ in the illuminated mid-thermosphere above 140 km. $N(^4S)$ can then be transported or mixed by molecular diffusion down into the lower thermosphere, where its reaction with NO (reaction Eq. 7.2) is the main loss process of NO. This chain of processes is summarized in Figure 10.

Our model experiments were carried out for solar minimum conditions, and this has an impact on the rate of formation of NO via reaction Eq. 2.1. As this reaction is strongly temperature dependent, higher temperatures in the mid-thermosphere during solar maximum would lead to higher values of NO, and less $N(^4S)$; consequently less downward propagation of $N(^4S)$ into the lower thermosphere, and a higher lifetime of NO there. In this sense, the mechanism described above and summarized in Figure 10 is likely more important during solar minimum conditions. Equally, the low auroral forcing at high latitudes during

early 2010 could contribute to the comparatively large impact of the thermospheric meridional circulation on the high-latitude lower thermosphere, as background values of both NO and $N(^4S)$ are than very low during polar night conditions, and the partitioning is likely more in favour of $N(^4S)$ than during geomagnetically active periods: formation of $N(^2D)$ in the lower thermosphere by continuing auroral activity would presumably lead to a larger ambient background of NO, and a higher ratio of NO to $N(^4S)$.

The mid-thermospheric formation of $N(^4S)$ is missing in models with their top below or near 140 km (WACCM-D, KASIMA). These models consider upper boundary conditions of both NO and $N(^4S)$, and the overestimation of NO in the low- and mid-latitude lower thermosphere in both models could indicate either an underestimation of the upper boundary value for $N(^4S)$ in these latitudes, or of the efficiency of the downward transport and mixing.

For models with their top in or above the mid-thermosphere (HAMMONIA, EMAC, WACCM-X) both a good representation

of the rate of $O_2$ photodissociation and a good representation of thermospheric transport and mixing are necessary for a realistic representation of lower thermospheric NO. This is particularly important for the enhanced NO layer in the polar winter lower thermosphere and upper mesosphere, which apppears to depend critically on the downward and poleward transport of $N(^4S)$ from its source regions in the mid- and low-latitude mid-thermosphere.





As the meridional circulation in the lower and middle thermosphere in the winter hemisphere appears to be significantly
affected by gravity waves, a better representation of the transport of gravity waves across the mesopause as well as the formation
of secondary and tertiary gravity waves is necessary to well represent NO in the polar winter lower thermosphere and upper
mesosphere, a prerequisite to realistic representation of the EPP indirect effect. This could be achieved, e.g., by models with
higher spatial resolution (Becker and Vadas, 2020; Liu et al., 2024a), or by gravity wave drag parameterizations focussing on
the thermosphere as described, e.g., in Miyoshi and Yiğit (2019).

*Author contributions.* MS, HL, TR, TS, and MES designed the experiments' setup and carried out the model experiments. MS analysed the
model experiments and wrote most of the manuscript. CA, SB and BF provided observational data. JMW provided the AISstorm particle
ionization rates. All authors contributed to discussion and interpretation of the results.

*Competing interests.* At least one of the co-authors is a member of the editorial board of Atmospheric Chemistry and Physics.

*Data availability.* SCIAMACHY NO in the MLT observation mode are available with a cc licence and can be accessed via https://www.
imk-asf.kit.edu/2939.php or via zenodo at https://zenodo.org/record/581253. MIPAS data can be obtained from the KITopen repository at
https://doi.org/10.5445/IR/1000156457 (Funke et al., 2023). COSMIC-1 data are available from the UCAR COSMIC Program under the doi
https://doi.org/10.5065/ZD80-KD74. Reprocessed level 2 electron densities were accessed on April 27, 2023. Postprocessed model results
and observational data as used in the figures will be published on the Radar4KIT repository at the time of publishing.

*Acknowledgements.* EMAC model experiments were performed on the HoreKa supercomputer funded by the Ministry of Science, Re-
search and the Arts Baden-Württemberg and by the German Federal Ministry of Education and Research. Simulations with HAMMO-
NIA have been performed on the ETH Zürich cluster EULER. TS acknowledges support from the Swiss National Science Foundation
(SNSF) project AEON (grant no. 200020E_219166) and Karbacher Fonds, Graubünden, Switzerland. MES acknowledges the Research
Council of Finland grant 335554-ICT-SUNVAC. The IAA team (BF and SB) acknowledges financial support from the Agencia Estatal de
Investigación, MCIN/AEI/10.13039/501100011033, through grant nos. PID2022-141216NB-I00 and CEX2021-001131-S. HLL acknowl-
edges support by the NCAR System for Integrated Modeling of the Atmosphere (SIMA) project. The authors acknowledge the NOAA
National Centers for Environmental Information (https://ngdc.noaa.gov/stp/satellite/poes/dataaccess.html) for the POES and Metop par-
ticle data used in AISstorm and give many thanks to the SuperMAG team (http://supermag.jhuapl.edu/) and their collaborators (http:
//supermag.jhuapl.edu/info/?page=acknowledgement). National Center for Atmospheric Research is a major facility sponsored by the Na-
tional Science Foundation under Cooperative Agreement No. 1852977. WACCM-X simulations were performed on NWSC/NCAR Cheyenne
Supercomputers with computing resources provided by the NCAR Strategic Capability (NSC) allocation and the Computational and Infor-
mation Systems Laboratory (CISL) at NCAR.



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
