# Peer review of "Thermospheric nitric oxide is modulated by the ratio of atomic to molecular oxygen and thermospheric dynamics during solar minimum"

_EGUsphere, 2024_

## Author Response (AR1)

**Dear Reviewer,**

Thank you for your concerns and suggestions, which will help focus the paper.

Based on the comments of both reviewers, we suggest the following main changes:

- Change title to "Thermospheric nitric oxide is modulated by the ratio of atomic to molecular oxygen and thermospheric dynamics during solar minimum"
- Move Figures 2 (NO timeseries) and 3 (electron densities) of the preprint to a supplement. Only the two extreme cases, WACCMx and EMAC, will be shown and discussed in Figures 5 and following. Figure 5 with all models will be moved to supplement as well. The discussion of the O/N2 ratio is also moved to the supplement, as it does not provide additional insights but strengthens the conclusions from the discussion of O/O2.
- Increase font sizes in all figures
- Add table listing advantages and disadvantages of different model geometries to Summary section

A more detailed response to your concerns is given below. Reviewer comments given in black, our response in blue.

This study focused on the simulation of NO in the lower thermosphere by comparing 5 numerical models with observations. They concluded that "two processes interacting with each other are identified as likely sources of these discrepancies, quenching of N(2D) by atomic oxygen in the mid-thermosphere, and meridional transport and mixing from the mid-thermosphere to the lower thermosphere". The results and conclusions will contribute to our knowledge on the variation of NO and also will contribute to further improve the first-principle based models in the future. However, there are some major issues to be addresses before it was considered to be published.

Here are some detailed concerns and some suggestions:

The structure of the paper lacks clarity, making it hard for readers to follow. I recommend having a native English speaker review and revise both the language and the logical flow to improve overall clarity.

Thank you for this suggestion. The paper was read carefully by co-author and native British speaker Dan Marsh before submission, and he will do this again before submission of a revised version.

The title of the manuscript is difficult to understand. Please consider rewriting it for clarity.

Thanks for pointing this out. We suggest to change the title to "Thermospheric nitric oxide is modulated by the ratio of atomic to molecular oxygen and thermospheric dynamics during solar minimum"

The font size of the text in the figures should be larger for better readability.

This will be addressed.

Figure 7: The authors did not discuss why O/O2[N2] from HAMMONIA is lower than that from WACCMx, because both of which considered photodissociation of O2 in the SRBC.

In HAMMONIA, total air density is lower than in WACCMx or EMAC above about 130 km, see figure of the snapshot along 0°E at 12:00 UTC on January 1, 2009 below. This has an impact on the reaction velocities of most reactions including quenching and photolysis reactions, potentially affecting the relative amounts of species. The reason why the density is lower in HAMMONIA was not explored further because it was felt that this is out of scope of the paper. Because we could not clarify this point to our complete satisfaction, we suggest to concentrate on the extreme cases WACCMx and EMAC in our analysis in a revised version.

I recommend the authors add a table to list the advantage and disadvantage of these models before the Summary section to clarify the simulation results.

That is a good suggestion, thank you! A table will be added:

| Top altitude  | 70-100 km
EMAC                                                                             | 115 – 150 km
KASIMA, WACCM-D                                                                               | >150 km
WACCMx, EMAC                                                            |
|---------------|-----------------------------------------------------------------------------------------------|---------------------------------------------------------------------------------------------------------------|------------------------------------------------------------------------------------|
| Advantages    | NOy upper boundary condition well constrained by observations, e.g., Sinnhuber et al., (2018) | Auroral NO source in model domain                                                                             | Auroral and EUV sources
of N and NO self-
consistently in model
domain    |
| Disadvantages | Source region of
thermospheric NO not
covered                                           | EUV production of N
above model top: upper
boundary condition
necessary, but not well
constrained | High spatial resolution necessary due to lack of adequate gw drag parameterization |

**Dear Reviewer.**

Thank you for your concerns and suggestions, which will help to focus the paper.

Based on the comments of both reviewers, we suggest the following main changes:

- Change title to "Thermospheric nitric oxide is modulated by the ratio of atomic to molecular oxygen and thermospheric dynamics during solar minimum"
- Move Figures 2 (NO timeseries) and 3 (electron densities) of the preprint to a supplement. Only the two extreme cases, WACCMx and EMAC, are shown in Figures 5 and following. Figure 5 with all models will be moved to supplement as well. The discussion of the O/N2 ratio is also moved to the supplement, as it does not provide additional insights but strengthens the conclusions from the discussion of O/O2.
- Increase font sizes in all figures
- Add table listing advantages and disadvantages of different model geometries to Summary section

A more detailed response to your concerns is given below. Reviewer comments given in black, our response in blue.

This pretty long paper utilized 6 models to check their ability to reproduce the climatology of NO. They also compare the model results with observations in electron density to test the difference in ionization. From my point of view, this, together with the NO comparison, is too much and oversized for a paper. I think a better revision can be carried out by only focusing the NO comparison, and maybe in more detail like NO during solar minimum quiet time and disturbed time. Then a following up paper can focus on the other comparisons.

We do appreciate that the paper is long. However, focusing only on the NO comparison does not make sense at this point in our opinion, as there already was a paper focusing on NO comparisons, Sinnhuber et al., 2022 referenced in the paper. The strong disagreement between different models in the lower thermosphere shown there are the motivation for this follow-up study. Not following up on why the NO differs so greatly from model to model would therefore be of little additional value compared to the previous study, and does not justify a standalone paper. To shorten and focus the paper, we suggest to move the model-observation intercomparison over the whole year (Fig. 2 of preprint) to supplementary material, and only very briefly summarize those results in the paper.

Also, for electron density comparison, I think it is not a good method to quantify why NO comparison has such difference. ....

Electron density stands for too much aspects and may not simply show the ionization.

We appreciate that electron densities can be difficult to interpret. However, both NO and electron densities can be considered indicators of atmospheric ionization, and considering this, we find it remarkable to note, and an important result, that modeled electron densities fall into a much narrower range, and agree much better with observations, than NO densities. However, for the sake of focusing the paper on the

explanations of the large variability of NO, we suggest to move this part to the supplementary material as well.

I think for NO, the author shall present the major terms that determine the NO density, then check these terms in detail.

The reactions governing formation and loss of NO in the lower thermosphere are summarized in the Introduction (Formation: Equations 1.1 to 6, Loss: Equations 7.1 and 7.2). We compared the rates of these reactions as implemented in WACCM and EMAC, and found that these did not differ significantly, with one exception: One significant difference between WACCMx and EMAC is the partitioning between N(2D) and N(4S) formation in the dissociation and dissociative ionization of N2 (Equations 1.1 and 1.2), which is given in Table 1 of preprint. This would favor a higher ratio of N(2D) to N(4S) in WACCMx than in EMAC. However, this is not what is observed comparison of  $N(^4S)$  and  $N(^2D)$  clearly shows much larger values of  $N(^4S)$  in WACCMx than in EMAC, while N(2D) values are comparable between both models (Figure 5 of preprint). The very high values of N(4S) lead to a significantly shorter NO lifetime in WACCMx compared to all other models (Figure 5 of preprint) due to the reaction of N + NO (Eq 7.2), and consequently, to the very low values of NO even though the rates of formation of NO might be comparable. The high values of  $N(^4S)$ can be explained by the high ratio of atomic oxygen to molecular oxygen as discussed in the paper, as  $N(^2D)$  is efficiently quenched to the ground state  $N(^4S)$  by atomic oxygen (Equation 3.1). We will clarify this point more in the revised version.

---

## Author Response (AR2)

Dear Editor,

Thank you for your positive assessment of the revised paper.

We have addressed your comments, response see below. As suggested, we went through the paper and found a few more issues, mostly typos; a list of all changes is also given below. Your comments in black, our response as well as the list of all changes in blue. Line numbers refer to the no-track changes version uploaded in May of this year. Post-processed model data are now uploaded to a public repository with a permanent identifier, and this information has been added to the data availability statement.

Kind regards,

Miriam Sinnhuber for all authors

The authors have responded appropriately to the reviewer's comments. The paper is now better organized and clearer. In going over the paper, I noticed a couple of minor typos (Line 160 - should be with upper and lower ...; Line 294 - add a space after NO).

Line 160: x-ray was changed to X-ray. This was changed throughout the manuscript, see list of all changes below.

Line 294: done.

I also noticed that there were several places where January 2009 is stated as the start of the model runs (Line 209 in the definition of the Long experiment; Line 100 of the supplement). I suspect these are also typos, but there is also mention of an initialization period in the paper. I suspect this is necessary since the Snapshot date is very close to January 2010 should that be the start of the run. Before publication, please clarify the dates of the model runs and describe the nature of the initialization period.

That's correct; most models did one year of spinup-time, but at least one model branched off from a multi-year experiment. Line 209: Changed "January 1, 2009" to "January 1, 2010" and added ", with at least one year of spinup-time before January 2010".

Line 100 in Supplementary Material: This was a typo, thanks for pointing this out. Changed to "January 9, 2010".

Thanks for an interesting paper.

Thank you!

**List of all Changes made:**

Changed x-ray to X-ray: lines 21, 32, 33, 160, 224, and line 32 of the Supplementary Material

Line 80: changed full stop after "MIPAS observations" to "as"

Lines 109, 130: changed "600 km" to "500 km", as this is the top altitude in the WACCM-X data-set of January 9, 2010.

Line 270: erased "in their common altitude range" as both models cover the altitude range shown.

Line 274: Changed "Up to 140 km" to "Above about 100 km"

**Data availability statement:**

Added "Ap and F10.7 used in Figure 1 are from the CMIP6 solar forcing data available at <a href="https://solarisheppa.kit.edu">https://solarisheppa.kit.edu</a>".

Last sentence changed to "Post-processed model results as used in the Figures of the main text and supplements are published on the repository Radar4KIT with a CC-bY-4 license and persistent identifier 10.35097/8b6tm3xvtxvgjtvb (https://radar.kit.edu/radar/en/dataset/8b6tm3xvtxvgitvb)."

**Supplementary material:**

Figures have been renamed S1, S2 etc, and all references in the main text and Supplementary material have been changed accordingly.